# Adversarially Robust Approximate Furthest Neighbor

**Kiarash Banihashem** [* 1]   **Jeff Giliberti** [* 1]   **Prashant Gokhale** [* 2]   **Samira Goudarzi** [* 1]
**MohammadTaghi Hajiaghayi** [* 1]   **Yuhao Liu** [* 2]   **Morteza Monemizadeh** [* 3]   **Sandeep Silwal** [* 2]

## Abstract

We work in the *adaptive query model*, where one is given a point set $P \subset \mathbb{R}^d$ and seeks to construct a data structure that can answer correctly and efficiently a sequence of *adaptive* queries. In this model, an adversary observes the answers returned by the data structure to previous queries $q_1, \ldots, q_{i-1}$ and, based on this information, chooses the next query point $q_i$. This setting captures strong forms of adaptivity that naturally arise in modern machine learning pipelines, and rules out many classical randomized techniques that assume oblivious queries. Our focus is the problem of *furthest neighbor search* in this adaptive setting, a fundamental problem in several learning tasks, including diversity maximization, outlier and anomaly detection, adversarial example generation, and more. We present the first adversarially robust data structure for $c$-approximate furthest neighbor queries that achieves query time $\tilde{O}(\min(dn^{1/c^2}, n^{2/c^2} + d))$. This matches the $n$ dependency in the query time of the seminal result by Indyk [SODA'03] for $c$-approximate furthest neighbor in the *oblivious setting*, and improves upon the $\tilde{O}(n + d)$ query time achieved via the adaptive distance estimation framework of Cherapanamjeri and Nelson [NeurIPS'20] for a wide range of natural parameters. To complement this result, we present an adversarial attack against oblivious approximate furthest neighbor algorithms. Specifically, we show that the data structure from the algorithm by Indyk fails to maintain its guarantees against adaptive queries.

[*]Equal contribution  [1]Department of Computer Science, University of Maryland, College Park, MD, USA  [2]Department of Computer Science, University of Wisconsin-Madison, Madison, WI, USA  [3]Department of Mathematics and Computer Science, TU Eindhoven, Eindhoven, The Netherlands. Correspondence to: Jeff Giliberti <jeffgili@umd.edu>.

*Proceedings of the 43ʳᵈ International Conference on Machine Learning*, Seoul, South Korea. PMLR 306, 2026. Copyright 2026 by the author(s).

## 1. Introduction

There has been increasing interest in understanding the behavior of algorithms when deployed in adaptive or adversarial environments. Such settings arise naturally in interactive data analysis, online learning, and security-sensitive applications, where future inputs may depend on past outputs of the algorithm. A broad line of work studies robustness and validity under adaptivity in areas including exploratory data analysis, statistical inference, and machine learning (Dwork et al., 2015a;b;c; Bassily et al., 2016; Dwork et al., 2017). These works focus primarily on preserving statistical guarantees when analysts adaptively explore a fixed dataset. Related concerns have also appeared in adversarial learning and security domains, including malware detection, intrusion detection, strategic classification, and autonomous systems (Biggio et al., 2013; Hardt et al., 2016; Goodfellow et al., 2014; Yuan et al., 2019; Liu et al., 2017; Papernot et al., 2016). In these applications, adaptivity can fundamentally invalidate guarantees of randomized algorithms designed for oblivious inputs.

Recent research has increasingly focused on the adversarial robustness of high-dimensional geometric algorithms. This body of work includes computing all distances from a query point to a dataset (Cherapanamjeri & Nelson, 2020), nearest neighbor search (Andoni et al., 2026), approximate diameter (Banihashem et al., 2025), clustering problems (Bateni et al., 2024; 2023), and other applications (Cherapanamjeri & Nelson, 2022a; Cherapanamjeri et al., 2023). Continuing this line of work, we investigate the adversarial robustness of sublinear algorithms for fundamental geometric primitives when faced with an adaptive adversary.

**Nearest and Furthest neighbors.** Nearest neighbor search is a fundamental computational primitive in high-dimensional data analysis and machine learning. It serves as a cornerstone for numerous applications including information retrieval, recommendation systems, metric learning, DNA sequencing, and more (Han et al., 2024; Shakhnarovich et al., 2006; Liu et al., 2023; Tagami, 2017). While the nearest neighbor problem has been studied extensively for decades, its dual counterpart—*furthest neighbor search*—has recently gained prominence due to its importance in modern machine learning workflows.

Furthest neighbor queries naturally arise in tasks such as outlier and anomaly detection (Muhr et al., 2023), diversity maximization (Anand et al., 2025), hard negative mining for contrastive learning (Robinson et al., 2021), adversarial example generation (Xiao et al., 2018), exploration in reinforcement learning, nonlinear dimensionality reduction (Vasiloglou et al., 2008; 2009), and complete linkage clustering (Defays, 1977; Dasgupta & Laber, 2024).

In many of these applications, queries are not predetermined but sequentially generated, with each query depending on previous responses. For example, in active learning or online exploration, the next query point is often chosen based on previously identified extreme or informative points. Similarly, in interactive data analysis and adaptive optimization, queries are refined based on intermediate outcomes. This sequential dependency fundamentally challenges classical randomized data structures, whose guarantees typically rely on the assumption that queries are fixed in advance.

To formalize this challenge, Cherapanamjeri & Nelson (2020) (and shortly thereafter (Ben-Eliezer et al., 2021)) introduced the *adaptive query model*. In this model, a data structure must answer a sequence of queries $q_1, q_2, \ldots$ where an adversary, after observing all previous answers, adaptively chooses the next query. This strong adversarial setting captures the feedback loops inherent in many machine learning pipelines and invalidates many classical randomized analyses. As shown in (Cherapanamjeri & Nelson, 2020), adaptivity can dramatically increase the complexity of even the most basic geometric queries.

**Our problem**   In this paper, we study *approximate furthest neighbor search* in the adaptive query model. Given a point set $P \subset \mathbb{R}^d$ of size $n$ and an approximation factor $c \geq 1$, the goal is to preprocess $P$ into a low-memory randomized data structure that, for each adaptively chosen query point $q$, returns a point $p \in P$ whose distance to $q$ is within a factor $c$ of the true furthest neighbor distance (see Section 2 for the formal setup). While exact furthest neighbor is computationally prohibitive in high dimensions, approximate solutions suffice for most learning applications and offer the potential for sublinear query time.

Prior to our work, the best known data structure for adaptive furthest neighbor queries was due to Cherapanamjeri & Nelson (2020) who achieved $\tilde{O}(n + d)$ query time for $c$-approximate queries, essentially matching a trivial linear scan over all points after a JL transformation (see e.g. (Johnson et al., 1984; Larsen & Nelson, 2017)). On the other hand, the best known *oblivious* algorithm by Indyk (2003) achieves $\tilde{O}(dn^{1/c^2})$ query time for a $c$-approximation. This leaves open a fundamental question:

*Is sublinear in $n$ query time achievable for approximate furthest neighbor search under full adaptivity?*

**Our contributions**   We answer this question affirmatively by designing a randomized sublinear data structure for *adaptive* approximate furthest neighbor search (AFN).

**Theorem 1.1.** *For any constants $c > 1$ and $\varepsilon > 0$, given a set $P$ of $n$ points, there is an adversarially robust algorithm for the approximate furthest neighbor (AFN) problem that returns a $c$-approximate-AFN in $\tilde{O}(dn^{1/c^2})$ query time with high probability, even if the queries are generated adaptively by an adversary, and using $\tilde{O}(d \cdot \min\{n, dn^{2/c^2}\})$ space.*

*Moreover, with query time $\tilde{O}(\min\{n^{2/c^2}, n\} + d)$, we can obtain a $c(1 + \varepsilon)$ approximation under the same adaptive model using $\tilde{O}(d^2 + d \cdot \min\{n, dn^{2/c^2}\})$ space.*

*Both variants have $\tilde{O}(d^2 n^{1+1/c})$ preprocessing time.*

Our result improves upon the $\tilde{O}(n + d)$ bound that can be achieved by combining the robust JL-like projection of Cherapanamjeri & Nelson (2020) with a linear scan over all the points for a wide range of natural parameters. For example, if $d = \text{poly}(\log n)$ then the first stated query time is sublinear in $n$ for all $c > 1$. If $c > \sqrt{2}$, then our second query time is sublinear in $n$ for all $d = o(n)$.

We complement our algorithmic result with an adversarial attack against the oblivious AFN data structure by Indyk (Indyk, 2003), demonstrating it fails under adaptive queries. Our attack exploits the underlying random projections by choosing the query point as a function of the projections. It artificially inflates certain projected distances to deceive the data structure into returning a point that is far from the true furthest neighbor, proving that a single, oblivious data structure is insufficient.

### 1.1. Techniques

At the heart of our adversarially robust solution is a three-step robustification. It constitutes a generic recipe that may appeal to other problems as well; we present it referring to generic data structures that solve the problem at hand.

1. **Base Data Structure**: An *oblivious* data structure $D$ that provides a "smooth" success guarantee for an arbitrary query point with sufficient, say constant, probability. That is, if we are ensured that the data structure can answer $c$-approximately for a query point $q$, the quality of the approximation should degrade smoothly for a query point $q'$ that is close to $q$.

   Because of this "smooth" requirement, we need to open up Indyk's algorithm (Indyk, 2003), based on random projections, to strengthen its guarantees. We introduce the concepts of *good projections* and *outlier projections* and use them to show that, for an arbitrary point $q$, one can achieve the same guarantees as Indyk's *and* an additional *slack guarantee*, with no asymptotic loss.

This slack guarantee will make neighbor points $q'$ of $q$ "good" deterministically, solely based on $q$ being good. This part appears in Section 2.1.

2. **Robust Data Structure**: A single randomized data structure may succeed on an individual query but fail on a nearby one chosen adversarially. To obtain guarantees that hold simultaneously for all possible queries, we ensure correctness on a representative set of query points, whose correctness guarantees extend to nearby points.

   We build a robust data structure $\mathcal{D} = D_1, \ldots, D_k$ composed of $k$ independent copies of $D$, each instantiated using fresh random bits. The parameter $k$ is chosen based on the success probability of $D$ for an arbitrary query point and, more importantly, the size of a fine covering of the *query space*; so that the final success probability will allow for a union bound over this covering.

   For *value* problems such as approximate distance estimation, it suffices to cover a unit ball around each point (Cherapanamjeri & Nelson, 2020). However, for *search* problems such as furthest neighbor, the identity of the returned point may change abruptly with the query location, and no a-priori bound without a dependency on the scale (i.e., the spread ratio of the input dataset) is available. A possible workaround used for approximate nearest neighbor is to consider a weak decision problem instead, where an answer should be returned only if there is a neighbor within a fixed radius $r$ (Andoni et al., 2026). Interestingly, none of these two approaches would work for AFN. In Section 2.2, we show how to construct a covering of the query space based only on the diameter of the dataset that yields a scale-free robustification.

3. **Efficient Querying**: To retrieve an answer, it is desirable not to query all of the $k$ base data structures. If a query can be answered correctly by an $\Omega(1)$-fraction of them, then querying $m = O(\log n)$ many (chosen uniformly at random) provides $m$ answers out of which one is correct with high probability. It remains to efficiently find a correct one among these. Since we get $\tilde{O}(n^{1/c^2})$ furthest neighbor candidates from our robust data structure, we can examine their distances in $\tilde{O}(d \cdot n^{1/c^2})$ time (which is our first guarantee).

   Improving the dependency on $d$ in $\tilde{O}(d \cdot n^{1/c^2})$ requires a more careful approach than a classical JL-transform. This is because a JL projection is, to the best of our knowledge, not resilient to adaptive queries. Our insight to avoid the $d$-factor is to gather a "small" set of FN candidates and use the robust distance estimation framework of (Cherapanamjeri & Nelson, 2020) to approximate their distances in $\tilde{O}(n^{2/c^2} + d)$ time. This, to our knowledge, is one of the first examples of how a robust algorithm can be used as a black-box in another robust

algorithm without worrying about the dependencies between input queries and intermediate output. This query process (and final algorithm) appears in Section 2.3.

We emphasize that the careful combination of the above steps is non trivial. It requires several new ideas to obtain an approximation quality and query time that match those of the best known oblivious algorithm, while providing robustness against an adaptive adversary *and* no dependency on the number of queries that our data structures can bear. This contrasts recent robust algorithms based on differential-private methods that rely on query-size dependent bounds (Andoni et al., 2026; Hassidim et al., 2022; Feng et al., 2025).

**Black-box vs. white-box adverserial models.** Our algorithm in fact works even against an adversary who is allowed to see the *internal randomness* of the algorithm. This is because we prove that, with high probability, we can correctly answer all queries, revealing the *past* randomness does not hurt our success guarantee. In the literature, this is usually denoted as a *white-box* adversary (Ajtai et al., 2022), in contrast to a *black-box* adversary which can only observe the outputted answers. Interestingly, this is the strongest guarantee one can hope for after determinism. In contrast, differential privacy-inspired approaches (see (Hassidim et al., 2022; Feng et al., 2025) and references therein) not only require extra query time to periodically re-build their data structures, but they also fall apart when information about their randomness leaks. This proves a *strict* separation between our framework and differential privacy-inspired frameworks. See also Section 1.2 for a more elaborate comparison.

### 1.2. Related Work

**Adversarial streaming model.** Many works have studied linear sketches and streaming algorithms in adversarial settings, often under assumptions quite different from the query model considered here. In the adversarial streaming model (Alon et al., 2021; Ben-Eliezer et al., 2022), the input is a sequence of adaptive updates and the goal is to approximate, collect, or compute some statistic while using as little space as possible. This setting has received a lot of attention since its formalization, with many works showing how to robustify classical streaming algorithms and estimation problems (see e.g. Braverman et al. (2021); Ben-Eliezer & Yogev (2020); Kaplan et al. (2021); Cohen et al. (2022); Ben-Eliezer et al. (2026); Lai & Bayraktar (2020); Woodruff & Zhou (2022); Cohen et al. (2023); Gribelyuk et al. (2024; 2025); Attias et al. (2024), and references therein). Closely related formalization of adaptivity are the white-box streaming model (Xiao et al., 2018; Ajtai et al., 2022; Feng et al., 2024), intermediate adversarial models (Sadigurschi et al., 2023), and adaptivity under insertions and deletions in the dynamic setting (Beimel et al., 2022).

**Adaptive data structures and sketches.** A major step towards formalizing adaptivity in geometric data structures was made by Cherapanamjeri & Nelson (2020), who introduced the *adaptive query model*. In this model, a randomized data structure must remain correct even when each query is chosen by an adversary that observes all previous answers. They showed that many standard Monte Carlo techniques fail under adaptivity, and developed new methods for approximate distance estimation that remain valid in this strong adversarial setting.

**Nearest neighbor search.** Nearest neighbor search (NNS) is a fundamental problem with extensive literature, motivated by applications in computer vision, information retrieval, and database systems (Bingham & Mannila, 2001; Shakhnarovich et al., 2008; Datar et al., 2004). Exact NNS suffers from prohibitive space complexity in high dimensions (e.g., Clarkson (1988); Meiser (1993)), leading to a long line of work on approximate nearest neighbor (ANN) search. Foundational results such as (Indyk & Motwani, 1998; Kushilevitz et al., 1998) achieve sublinear query time using locality-sensitive hashing.

However, classical ANN data structures assume that queries are fixed in advance. Under adaptive queries, their guarantees can fail. Recent work on Las Vegas variants of ANN (Ahle, 2017; Wei, 2022; Pagh, 2016; Sankowski & Wygocki, 2017) ensures correctness for all queries, but only guarantee efficient query time when queries are non-adaptive. Algorithms that explicitly support adaptive nearest neighbor queries (e.g., Kleinberg (1997); Kushilevitz et al. (1998)) achieve sublinear query time at the cost of very large space usage, ranging from $\Omega(n^d)$ to $\Omega(n^{O(1/\varepsilon^2)})$.

**Distance estimation and nonparametric methods.** Approximate distance estimation (ADE) is a more general primitive than nearest neighbor search and plays a key role in nonparametric estimation. Many learning algorithms, including kernel regression, support vector machines, and density estimation, require distance information to a large fraction of the dataset rather than to a single nearest neighbor (e.g., (Wand & Jones, 1994; Hofmann et al., 2008; Altman, 1992; Simonoff, 2012; Atkeson et al., 1997)). In such settings, restricting attention to a small number of neighbors may be insufficient, and the appropriate number of neighbors may depend on the query point itself.

Cherapanamjeri and Nelson (Cherapanamjeri & Nelson, 2020) developed the first adaptive data structures for ADE, providing $(1 + \varepsilon)$-approximate distance estimates to all points with near-linear query time. Their framework however does not yield sublinear-time algorithms for extremal queries such as nearest or furthest neighbor search.

**Furthest neighbor search.** Compared to nearest neighbor search, furthest neighbor search has received significantly less attention, despite its importance in many computational tasks. In high dimensions, exact furthest neighbor search is computationally expensive, and even approximate variants are challenging under adaptivity. Most relevant to our work, is the oblivious $c$-approximate FN algorithm by (Indyk, 2003) with $\tilde{O}(n^{1/c^2+d})$ query time[1] and the use of the ADE framework of (Cherapanamjeri & Nelson, 2020) to answer adaptive furthest neighbor queries in $\tilde{O}(n + d)$ time. Whether sublinear query time is achievable for AFN under full adaptivity has remained an important open question.

## 2. A Robust AFN Algorithm

**Formal problem definition and model.** Let $P \subset \mathbb{R}^d$ be a fixed set of $n$ points. The $c$-approximate furthest neighbor problem is to maintain a data structure that, for any query $q \in \mathbb{R}^d$, returns a point $p \in P$ satisfying $\|p - q\| \geq \frac{1}{c} \max_{x \in P} \|x - q\|$.

In the *adaptive adversarial* setting, an adversary chooses a sequence of queries $q_1, q_2, \ldots$ adaptively, where each $q_t$ may depend on all previous answers returned. As intuition for why the adversarial setting is challenging, note that if the adversary can construct subsequent queries that exploit "blind spots" (e.g., orthogonal directions) in the data structure, the runtime and approximation guarantees may fail to hold. We require that the algorithm answers all queries correctly with high probability over its internal randomness, regardless of the adaptive nature of the queries, and that its time and space bounds hold deterministically.

| Parameter | Value | Meaning |
|---|---|---|
| $c$ | arbitrary | approximation factor |
| $\epsilon$ | arbitrary | projection error |
| $\delta$ | $1/n$ | slack parameter |
| $\Delta$ | $\max_{p,p' \in P} \|p - p'\|$ | diameter |
| $t$ | $\Theta(\sqrt{\log n})$ | technical threshold |
| $k$ | $\tilde{\Theta}(d)$ | # of base DS |
| $N$ | $\tilde{\Theta}(n^{1/c^2})$ | # of proj. per DS |
| $m$ | $\Theta(\lg n)$ | # of samples |

*Table 1.* Algorithm parameters.

There are a number of global parameters that we will use in our algorithm; we provide Table 1 for ease of reference.

---

[1] The $c$-approximation in (Indyk, 2003) is achieved after using a $(1 + \varepsilon)$-JL transformation, thus the actual approximation factor is $(1 + \varepsilon)c$, unless the query time is increased to $\tilde{O}(n^{1/c^2}d)$. We make this distinction explicit in our results.

## 2.1. Base Data Structure

The fundamental units of our construction are random projections. Let $a$ be a $d$-dimensional vector with each coordinate chosen randomly and independently from a normal distribution $\mathcal{N}(0, 1)$. For a query $q$ and a dataset $P \in \mathbb{R}^{d \times n}$, if $p^* \in P$ is the furthest neighbor of $q$, the value $|a \cdot p^* - a \cdot q|$ is likely to be large, likewise, the projection of any other point $p' \in P$ that is closer to $q$ than $p^*$, $|a \cdot p' - a \cdot q|$, is likely to be smaller. If this is the case, we say that the query $q$ is *good* with respect to $a$. Concretely, our definition of "goodness" for a query point is more general: (1) it asserts the optimality of the solution that we can find for $q$ in terms of the final approximation factor $c$; and, (2) it introduces some *slack* to ensure that if $q$ is good, so is any other query point $q'$ close to $q$. The parameter $\delta := 1/n$ measures the slack and will not affect results asymptotically.

**Definition 2.1** ($q$ is $(c, \delta)$-good). *Given dataset $P \in \mathbb{R}^{d \times n}$, a query point $q \in \mathbb{R}^d$, a set $A \in \mathbb{R}^{d \times N}$ of projection vectors, and a parameter $t \geq 1$, the point $q$ is $(c, \delta)$-good for $A$ if the following two properties hold for some $c > 1, \delta > 0$. Let $p^*$ be the furthest neighbor of $q$ in $P$.*

1. *There is a* good projection *of the pair $(q, p^*)$, that is, there exists $a \in A$ such that*

$$a \cdot p^* - a \cdot q \geq t \|p^* - q\| \frac{1 + \delta}{c}.$$

2. *The set of* outlier projections *$B_{q,A}^{c,\delta}$ has size at most $8N$, where*

$$B_{q,A}^{c,\delta} := \{(p', a) \in P \times A : \frac{\|p' - q\|}{\|p^* - q\|} < \frac{1 + \delta}{c},$$
$$a \cdot p' - a \cdot q \geq t \|p^* - q\| \frac{1 - \delta}{c}\}$$

Note that if a point is $(c, \delta)$-good, then it is also $(c, \delta')$-good for any $\delta' \leq \delta$.

Our goal now is to construct a random projection matrix $A$ that ensures that, with constant probability, an arbitrary point $q$ is good for some suitable parameters, while minimizing the size of $A$.

We first state two known results about projecting a point onto a random Gaussian vector.

**Claim 2.2** (cf. Claims 2 and 3 of (Indyk, 2003)). *Given a random projection vector $a$ with $a_j \sim \mathcal{N}(0, 1)$, a query $q$, and a pair of points $p$ and $p'$ such that $\frac{\|p' - q\|}{\|p - q\|} < \frac{1 + \delta}{c}$ for any $c > 1, \delta \in (0, 1/2)$, then we have*

$$\Pr_a \left[ a \cdot p - a \cdot q \geq \frac{t \|p - q\|}{c}(1 + \delta) \right] \geq \frac{1}{t} \cdot n^{-\frac{1 + O(\delta)}{c^2}}, \tag{1}$$

$$\Pr_a \left[ a \cdot p' - a \cdot q \geq \frac{t \|p - q\|}{c}(1 - \delta) \right] \leq \frac{1}{n}, \tag{2}$$

*for $t = \Theta(\sqrt{\log n})$ that is the solution to $e^{t^2 \frac{(1 - \delta)^2}{2(1 + \delta)^2}}/t = 2n$.*

The construction of (Indyk, 2003), later refined by (Pagh et al., 2015), uses a projection matrix of size $\Theta(n^{1/c^2} \sqrt{\log n})$ to ensure that a query $q$ is $(c, 0)$-good. We give a more general version of their result below.

**Lemma 2.3.** *For a query $q$, if $A \in \mathbb{R}^{d \times N}$ consists of $N$ independent random projection vectors with each coordinate distributed as $\mathcal{N}(0, 1)$, with $N = \Theta(n^{(1 + O(\delta))/c^2} \sqrt{\log n})$, then $q$ is $(c, \delta)$-good with probability at least $3/4$, for any desired $c, \delta \in (0, 1/2)$.*

*Proof.* Observe that property (1) of Def. 2.1 does *not* hold with probability at most

$$(1 - 1/t \cdot n^{-\frac{(1 + O(\delta))}{c^2}})^N \leq 1/8,$$

by Equation 1 and the choice of $N$. Moreover, from Equation 2, the expected total number of outlier projections is at most $N$. By Markov's inequality, this quantity is greater than $8N$ with probability at most $1/8$. Observe that points at distance at least $\|q - p\| \frac{(1 + \delta)}{c}$ do not hurt the probability of being good. Therefore, property (2) of Def. 2.1 fails to hold with probability at most $1/8$. With probability at least $3/4$, we can guarantee that $q$ is $(c, \delta)$-good for such $A$. $\square$

Let $\Delta$ be the diameter of $P$ (cf. Table 1). Let $q$ be $(c, \delta)$-good for a fixed $A$. We now prove that if a query $q'$ is sufficiently close to $q$ in terms of $\Delta$ and $n$, then $q'$ is $c$-good; note we omit the parameter $\delta$ when a point is $(c, 0)$-good.

**Lemma 2.4** ($q'$ has a good neighbor). *Given a pair of query points $q, q' \in \mathbb{R}^d$ such that $\|q - q'\| \leq \frac{\Delta}{n^3}$, if $q$ is $(c, \delta)$-good for $\delta \geq 1/n$ and a projection matrix $A$ such that for every $a \in A$, $\|a\| \leq n$, then $q'$ is $(c, 0)$-good.*

*Proof.* Let $p^*$ be the furthest neighbor of $q$, and $\tilde{p}^*$ be the furthest neighbor of $q'$. We prove both properties in turn.

**Property 1** (*Good Projection*): Let $p \in P$ and $a \in A$ be the point-vector pair satisfying property (1) for $q$. We show it satisfies property (1) for $q'$ with $\delta = 0$. First, observe that

$$a \cdot p - a \cdot q' = (a \cdot p - a \cdot q) + a \cdot (q - q')$$
$$\geq \frac{t \|p^* - q\|}{c}(1 + \delta) - \|a\| \|q - q'\|$$
$$\geq \frac{t \|p^* - q\|}{c}(1 + \delta) - 2 \frac{\|p^* - q\|}{n^2}$$
$$= \frac{t \|p^* - q\|}{c} \left(1 + \delta - 2 \frac{c}{tn^2}\right),$$

where in the second inequality we used the fact that $\|a\| \leq n$ (see Def. 2.1) and that $\|p^* - q\| \geq \Delta/2$, since $p^*$ is its

furthest neighbor. We now relate $\|p^* - q\|$ with $\|\tilde{p}^* - q'\|$:

$$
\begin{aligned}
\|p^* - q\| &\geq \|\tilde{p}^* - q\| \\
&\geq \|\tilde{p}^* - q'\| - \|q - q'\| \\
&\geq \|\tilde{p}^* - q'\|(1 - 2/n^3).
\end{aligned}
$$

Noticing that $(1 + \delta - 2c/(tn^2))(1 - 2/n^3) \geq 1$ for $\delta \geq 1/n$ yields the claimed bound.

**Property 2** *(Outlier Projections)* We show that $B_{q',A}^{c,0} \subseteq B_{q,A}^{c,\delta}$. Recall that a necessary condition for $p'$ to be an outlier for $q'$ is that $\frac{\|p' - q'\|}{\|\tilde{p}^* - q'\|} < \frac{1}{c}$. We first show that the above implies that

$$
\frac{\|p' - q\|}{\|p^* - q\|} \leq \frac{(1 + 1/\delta)}{c}.
$$

Thus, we have

$$
\begin{aligned}
\|q - p'\| &\leq \|q - q'\| + \|p' - q'\| \\
&< \|q - q'\| + \frac{\|\tilde{p}^* - q'\|}{c} \\
&\leq \|q - q'\| \left(1 + \frac{1}{c}\right) + \frac{\|\tilde{p}^* - q\|}{c} \\
&\leq \left(1 + \frac{8}{n^3}\right) \frac{\|p^* - q\|}{c},
\end{aligned}
$$

which proves the claim since $\delta \geq 1/n$.

It remains to show that if $a \cdot p' - a \cdot q' \geq \frac{t\|\tilde{p}^* - q'\|}{c}$ holds, then $a \cdot p' - a \cdot q \geq \frac{t\|p^* - q\|}{c}$ also holds and, thus, $(a, p')$ must be an outlier for $q$. We have

$$
\begin{aligned}
a \cdot p' - a \cdot q &= (a \cdot p' - a \cdot q') + a \cdot (q' - q) \\
&\geq \frac{t\|\tilde{p}^* - q'\|}{c} - \|a\|\|q' - q\| \\
&\geq \frac{t\|p^* - q'\|}{c} - \|a\|\|q' - q\| \\
&\geq \frac{t\|p^* - q\|}{c} - \|q' - q\|(\|a\| + t) \\
&\geq \frac{t\|p^* - q\|}{c} \left(1 - \frac{2(\|a\| + t)}{n^3}\right),
\end{aligned}
$$

where the last term is greater than $(1 - \delta)$. $\qquad\square$

We are now ready to construct the oblivious data structure that our robust algorithm will rely on.

**Lemma 2.5** (Base Data Structure). *For a set of $n$ points $P$ in $\mathbb{R}^d$, a parameter $c > 1$, and a projection matrix $A \in \mathbb{R}^{d \times N}$, one can construct a data structure $D$ that when queried with a point $q$ that is $c$-good for $A$, it returns a set $S$ of $O(N)$ points that contain a $c$-furthest neighbor of $q$. The data structure $D$ has $O(dN \log n)$ query time, uses $\tilde{O}(dN^2)$ space, and takes $O(dNn \log n)$ time to be constructed.*

*Proof.* Note that $A$ consists of $N$ $d$-dimensional vectors $A = \{a_1, \ldots, a_N\}$. We store the scalar projections $\{a_i \cdot p \mid p \in P\}_{i \in [N]}$ in $N$ sorted lists, $\mathcal{L} = \{L_1, \ldots, L_N\}$. These lists can be constructed in time $O(N \cdot dn \log n))$ and, for each list, only the *largest* $O(N)$ projections will be used, so the total space can be reduced to $\tilde{O}(dN^2)$.

Our approach is to take the largest $8N + 1$ candidates in terms of $(a \cdot p - a \cdot q)$ across all sorted lists. Since lists are sorted by $a_i \cdot p$, we can first build a max-heap of $N$ elements, with the largest entry from each list and key-value $\langle a_i \cdot p - a_i \cdot q, i \rangle$. Then, we repeatedly pop the top element from the heap, and insert the next largest element from the same list (note that we stored such index in the value of the heap entry). Within $O(N \log n + d)$ time, we can find the top $8N + 1$ candidates. Among them, we can show that there is a $c$-AFN using the fact that $q$ is $c$-good for $A$, i.e.

$$
\max_{p \in \mathcal{L}[8N+1]} \|q - p\| \geq \|q - p^*\|/c.
$$

Recall that by property (1) of Def. 2.1, there is a good projection vector $a$ such that $a \cdot p^* - a \cdot q \geq t\|p^* - q\|/c$. Moreover, the number of outlier projections greater than $t\|p^* - q\|/c$ is at most $8N$ by property (2) of Def. 2.1. Therefore, among the $8N + 1$ returned points there is at least one $p$ that is a $c$-AFN. $\qquad\square$

### 2.2. Robustification via Union Bound

A single random projection matrix offers limited guarantees. To handle adaptive queries, we construct a robust data structure that succeeds for *all* possible queries with high probability. Let $D$ be the base data structure from Lemma 2.5. Our robust data structure $\mathcal{D} = D_1, \ldots, D_k$ is composed of $k$ independent copies of $D$. To achieve a "for-all" guarantee, we will choose $k$ large enough to union bound over a suitable covering of the query space.

For a set of $n$ points $P$ in $\mathbb{R}^d$, we define the center of $P$ as $ct(P)$ and the width of a bounding box around $P$ as $bw(P)$ (see Definition B.1). Without loss of generality, we can assume that $ct(P) = \mathbf{0} \in \mathbb{R}^d$.

Let $R = \frac{(1+c)\sqrt{d}}{2(c-1)} bw(P)$. By the definition of $bw(P)$, we can upper and lower bound the distance between the query $q$ and any point $p \in P$ (cf. Lemma B.2). This will help us show that it suffices to restrict queries to a ball of radius $R$ centered at $ct(P)$, $B = B(ct(P), R)$: queries outside this ball can be trivially answered.

**Lemma 2.6** (trivial query). *For a set of $n$ points $P$ in $\mathbb{R}^d$ and any query $q$ such that $\|q - ct(P)\| \geq R$, we have*

$$
\min_{p' \in P} \|p' - q\| \geq \frac{\max_{p \in P} \|p - q\|}{c}.
$$

Our goal is now to prove that any other query inside of $B(0, R)$ is $c$-good for a constant fraction of our base data

structures. For the purpose of the analysis, we build a grid $Q$ covering $B(0, R)$ with a suitably small spacing determined later (see Def. B.3 for a formal definition of a grid). This ball clearly includes any query for which a trivial answer does not suffice (cf. Lemma 2.6). Our strategy is to ensure that a grid point is $(c, 1/\delta)$-good, so that any other point $q'$, which $q$ is sufficiently close to, will be $c$-good.

The next lemma establishes a key relationship between the no. of base data structures and the no. of grid points.

**Lemma 2.7.** *Let* $\mathcal{A} = \{A_1, \dots, A_k\}$ *be a set of independent random projection matrices, each associated with a base data structure* $D_i \in \mathcal{D}$ *and consisting of* $N = \Theta(n^{(1+O(\delta))/c^2} \sqrt{\log n})$ *independent random projection vectors with coordinates* $\sim \mathcal{N}(0, 1)$. *For a set of points* $Q$ *in* $\mathbb{R}^d$, *every point* $q \in Q$ *is* $(c, \delta)$-*good for at least* $k/2$ *many projection matrices with probability at least* $1 - 1/n^3$ *for* $k = \Theta(\log(|Q|n))$ *and any* $c, \delta \in (0, 1/2)$.

*Proof.* Let $A(q) = \{i \in [k] \mid q \text{ is } (c, \delta)\text{-good for } A_i\}$. We consider the following event:

$$E = \{\exists q \in Q : |A(q)| < k/2\}.$$

and need to show that $\Pr[\bar{E}] \geq 1 - 1/n^3$.

Consider a fixed $q$ and let $\{X_i\}_{i \in [k]}$ be indicator random variables for the event that $q$ is $(c, \delta)$-good for $A_i$. Let $X = \sum_i X_i$. By Lemma 2.3, we know that $\Pr[X_i] \geq 3/4$ for any $i$, and thus $\mathbb{E}[X] \geq 3k/4$. Since the $A_i$'s are independent, we can apply a Chernoff bound to show that $X$ is sharply concentrated around its expectation:

$$\Pr\left[X < k/2\right] \leq \Pr\left[X < \left(1 - \frac{1}{3}\right) \cdot \mathbb{E}[X]\right]$$
$$= e^{-\Omega(k)}.$$

By taking a union bound over the set $Q$, $\Pr[\bar{E}] \geq 1 - |Q|e^{-\Omega(k)} \geq 1 - 1/n^3$, for $k = \Theta(\log(n|Q|))$. $\square$

Recall that Lemma 2.4 establishes a connection between the $(c, \delta)$-goodness of a query point $q$ and the $c$-goodness of a neighbor query within distance at most $\Delta/n^3$ in $P$. We use this connection to prove that all points in the query space covered by the grid $Q$ are $c$-good with high probability.

**Lemma 2.8.** *Let* $\mathcal{A} = \{A_1, \dots, A_k\}$ *be projection matrices such that every vector* $a \in A_i$ *satisfies* $\|a\| \leq n$ *for all* $i \in [k]$. *Let* $Q = G_{\eta, r}$ *be a grid over the ball* $B(0, r)$ *with spacing* $\eta = \frac{bw(P)}{\sqrt{d}n^3}$.

*If every point* $q \in Q$ *is* $(c, \delta)$-*good for at least* $k/2$ *projection matrices, then every point* $q' \in B(0, r)$ *is also* $c$-*good for at least* $k/2$ *projection matrices, for* $\delta \geq 1/n$ *and any* $c > 1$.

*Proof of Lemma 2.8.* Consider an arbitrary point $q' \in B(0, r)$. If $q' \in Q$, the claim follows trivially. Else, let

$q \in Q$ be the nearest grid point of $q'$. By Lemma B.4 and the fact that $bw(P) \leq \Delta$, we have that $\|q - q'\| \leq \eta\sqrt{d} \leq \frac{bw(P)}{n^3} \leq \frac{\Delta}{n^3}$. Therefore, we can apply Lemma 2.4, that is, if $q$ is $(c, \delta)$-good for a matrix $A$, then $q'$ is $c$-good for the same $A$, provided that $\delta \geq 1/n$. Since $q$ is $(c, \delta)$-good for at least $k/2$ many projection matrices, so is $q'$. $\square$

## 2.3. Efficient Querying and Final Construction

In this section, we first discuss how to query our $k$ independent base data structures efficiently, and then present our final algorithm.

The next lemma proves that querying $O(\log n)$ base data structures is enough to find an approximate answer with high probability.

**Lemma 2.9.** *Let* $m = \Theta(\log n)$ *and* $q$ *be a query point that is* $c$-*good for at least* $k/2$ *matrices out of* $A_1, \dots, A_k$. *Let* $i_1, \dots, i_m$ *be indices sampled uniformly at random with repetitions from* $\{1, \dots, k\}$. *With high probability, there is an index* $i_j$ *such that* $q$ *is* $c$-*good for* $A_{i_j}$ *for* $i_j \in \{i_1, \dots, i_m\}$.

*Proof.* Let $S = \{i \in [k] \mid q \text{ is } c\text{-good for } A_i\}$. By assumption, $|S| \geq k/2$. Thus, a uniformly random index from $\{1, \dots, k\}$ lies in $S$ with probability at least $1/2$. Because the indices $i_1, \dots, i_m$ are sampled uniformly at random with repetitions from $\{1, \dots, k\}$, the probability that none of them lies in $S$ is at most $(1/2)^m = 2^{-m}$. Therefore, by choosing $m = \Theta(\log n)$, with high probability, $q$ is $c$-good for one of the matrices. $\square$

We make use of the adaptive distance estimation framework by Cherapanamjeri & Nelson (2020) to efficiently process our candidate answers.

**Lemma 2.10.** *[(Cherapanamjeri & Nelson, 2020; 2022b)]* *Given a point set* $P'$ *of* $n'$ *points and a parameter* $\varepsilon > 0$, *there is a robust data structure that given a query* $q$ *and a subset* $\hat{P} \subseteq P'$, *for every* $x_i \in \hat{P}$ *it returns an estimate distance* $\tilde{d}_i$ *such that*

$$(1 - \varepsilon)\|q - x_i\| \leq \tilde{d}_i \leq (1 + \varepsilon)\|q - x_i\|.$$

*in* $\tilde{O}(|\hat{P}| + d)$ *query time. It uses* $\tilde{O}(d(n' + d))$ *space and* $\tilde{O}(n'd)$ *preprocessing time.*

We are now ready to prove our main result.

**Theorem 1.1.** *For any constants* $c > 1$ *and* $\varepsilon > 0$, *given a set* $P$ *of* $n$ *points, there is an adversarially robust algorithm for the approximate furthest neighbor (AFN) problem that returns a* $c$-*approximate-AFN in* $\tilde{O}(dn^{1/c^2})$ *query time with high probability, even if the queries are generated adaptively by an adversary, and using* $\tilde{O}(d \cdot \min\{n, dn^{2/c^2}\})$ *space.*

*Moreover, with query time* $\tilde{O}(\min\{n^{2/c^2}, n\} + d)$, *we can obtain a* $c(1 + \varepsilon)$ *approximation under the same adaptive model using* $\tilde{O}(d^2 + d \cdot \min\{n, dn^{2/c^2}\})$ *space.*

*Both variants have $\tilde{O}(d^2 n^{1+1/c})$ preprocessing time.*

*Proof.* We instantiate $k$ base data structures $\mathcal{D} = \{D_1, \ldots, D_k\}$, each defined by a random and independent projection matrix $A_i \in \mathbb{R}^{d \times N}$ with each coordinate distributed as $\mathcal{N}(0,1)$, for $k = \Theta(d \log(dn/(c-1)))$ and $N = \Theta(n^{(1+O(\delta))/c^2} \sqrt{\log n})$. As a technical condition, we require that each vector $a$ satisfies $\|a\| < n$ for $a \in \cup_i A_i$. To this end, we use the standard bound on the norm of a random Gaussian vector: $\Pr[\|a\| \geq n] \leq de^{-n^2/2}$ (see e.g. (Feller, 1991)). Thus, with probability at least $1 - kNde^{-n^2/2} \gg 1 - 1/\text{poly}(n)$ this condition holds.

Let $n' := \tilde{O}(\min\{n, kN^2\})$. By Lemma 2.5, $\mathcal{D}$ can be constructed in $\tilde{O}(kdNn)$ time and uses $\tilde{O}(kN^2 + dn')$ space. Note that there at most $n'$ points (out of all $n$ points) that can be return by $\mathcal{D}$ as furthest neighbor candidates. Therefore, we construct the robust distance estimation data structure from Lemma 2.10 solely on this subset of $n'$ points, which takes $\tilde{O}(dn')$ preprocessing time, and $\tilde{O}(d+n')$ space. Plugging in the values of $k$ and $N$ yields the claimed preprocessing and space bounds. See Algorithm 1 for the pseudocode.

---

**Algorithm 1** Robust Furthest Neighbor Preprocessing

---

1: **Input:** $P \in \mathbb{R}^{d \times n}$, parameters $c, \varepsilon > 0$.
2: $N = \Theta\left(n^{1/c^2 + O(\delta)} \sqrt{\log n}\right), \delta = 1/n$.
3: $k = \Theta(d \log(dn/(c-1)))$.
4: Initialize $\mathcal{D} = \{D_1, \ldots, D_k\}$ with associated matrices $A_1, A_2, \ldots, A_k$ where each $A_i \in \mathbb{R}^{d \times N}$, and each entry $a_{ij} \sim \mathcal{N}(0,1)$.
5: $L_{ij} \leftarrow \emptyset$ for $i \in [k], j \in [N]$.
6: **for** $i = 1$ to $k$ **do**
7:     **for** $j = 1$ to $N$ **do**
8:         $L_{ij} \leftarrow \{a_{ij} \cdot p \mid p \in P\}$.
9:         Retain only $8N+1$ largest projections in $L_{ij}$.
10:     $L_i \leftarrow \bigcup_{j=1}^{N} L_{ij}$
11: $\hat{P} \leftarrow \{p \in P \mid a_{ij} \cdot p \in L_{i,j} \text{ for some } i \in [k], j \in [N]\}$
12: Build data structure $\mathcal{S}$ on $\hat{P}$ using Lemma 2.10.

---

Next, we show correctness for *all* queries with high probability. That is, we prove that all queries are $c$-good for at least $k/2$ base data structures with high probability. Recall that $R = \frac{(1+c)\sqrt{d}}{2(c-1)} bw(P)$. Consider first a query $q^{out}$ outside of $B(0, R)$. Here, we have $\|q^{out}\| > \frac{(1+c)\sqrt{d}}{2(c-1)} bw(P)$. Therefore, by Lemma 2.6, $q^{out}$ is $c$-good for any $A$.

Consider now query points in $B(0, R)$. Let $Q = G_{\eta, r}$ be a grid over the ball $B(0, R)$ with spacing $\eta = bw(P)/\sqrt{d}n^3$. By Lemma B.4, the number of points in $Q$ is bounded by $O((R/\eta)^d) = O((dn^3/(c-1))^d)$. Thus, by our choice of $k$ and Lemma 2.7, every $q \in Q$ is $(c, \delta)$-good for at least

$k/2$ projection matrices with probability at least $1 - 1/n^3$. Thus, each query point in $B(0, R)$ is $c$-good by Lemma 2.8.

---

**Algorithm 2** Furthest Neighbor Query

---

1: **Input:** Query point $q \in \mathbb{R}^d$
2: $m = \Theta(\log n)$.
3: Choose random indices $I = \{i_1, \ldots, i_m\} \subseteq [k]^m$.
4: $C \leftarrow \emptyset$         ▷ Furthest neighbor candidates
5: **for** each $i \in I$ **do**
6:     $C_i \leftarrow 8N+1$ largest projections in $L_i$ sorted by $a_{ij} \cdot p - a_{ij} \cdot q$ for $j \in [N]$.
7: $C \leftarrow \bigcup_{i \in I} C_i$.
8: Use $\mathcal{S}$ to find $\hat{p}$ such that $\|\hat{p} - q\| \geq \frac{\max_{p \in C} \|p - q\|}{1 + \varepsilon}$
9: **return** $\hat{p}$

---

When a query $q$ comes, we can apply Lemma 2.9 to pick $m = \Theta(\log n)$ random base data structures to answer: with high probability, among these, there is at least one projection matrix for which $q$ is $c$-good. Here, it is important to note that, since we use fresh randomness to choose $m$ indices, this process is not affected by the adaptivity of the adversary. Now there are two ways to proceed. First, we can project the query point to each of the $m$ datastructures and then use the guarantees of each of these base datastructures from Lemma 2.5. This results in the query time of $\tilde{O}(dn^{1/c^2})$ and returns a $c$ approximation.

Otherwise, we know that each of the $m$ base datastructures only have $O(N^2)$ points stored. Let $C$ be this candidate subset. We use the data structure from Lemma 2.10 to estimate all distances between $C$ and $q$ up to a $(1+\varepsilon)$-factor with high probability, and then pick $\hat{p}$ such that $\|\hat{p} - q\| \geq \max_{p \in C} \frac{\|p - q\|}{1 + \varepsilon}$. Since our $c$-good guarantee ensures that $C$ contains a $c$-approximate neighbor by Lemma 2.5, the final approximation $(1+\varepsilon)c$ follows, and the query time is $\tilde{O}(\min\{N^2, n\} + d)$. Since $\delta = 1/n$, the term $n^{O(\delta)} = O(1)$ and the claimed bounds follow. Algorithm 2 illustrates this process. □

## 3. Adaptive query against oblivious algorithms

In this section, we devise an adversarial attack against the oblivious AFN data structure by (Indyk, 2003), which uses random projections to find an AFN. Our attack works even in the setting where no changes to the underlying dataset are allowed, but only adaptive queries.

We start by recalling the underlying principle behind the base data structure, and why it fails. It relies on the fact that for a vector $v$ satisfying $\|v\|_2 = 1$, if we take its inner product with a random vector $a$ sampled from the standard Gaussian distribution $\mathcal{N}(0,1)$, then there exists $t \in \Theta(\sqrt{\log N})$ such that the following hold:

1. $\langle a, v \rangle$ is at least $t/c$ with probability at least $n^{-1/c^2}/t$.

2. $\langle a, v \rangle$ is more than $t$ with probability at most $1/n$.

We can show that these properties break when the choice of $v$ depends on $a$. Let $v$ be in the same direction as $a$, i.e., $v = a/\|a\|_2$. One can show that $\Pr[\|a\|_2 \leq \sqrt{d}/2] \leq e^{-9nd/64} = e^{-\Theta(nd)}$. Assuming $\|a\|_2 \geq \sqrt{d}/2$, the inner product $\langle a, v \rangle$ is at least $\sqrt{d}/2$ which is larger than $t$ for $d \geq \omega(\sqrt{\log n})$. As such, this construction breaks the main principle behind the algorithm.

We next describe the specific algorithm for which we show a failure result. The algorithm instantiates $N$ random vectors $a_1, \ldots, a_N$, each sampled from $\mathcal{N}(0, 1)$, and calculates inner products $\langle a_i, q - p \rangle$ for all $i \in N, p \in P$. It sorts these inner products by absolute value and, for the largest (in absolute value) $O(N)$ inner products, it checks the distance between $q$ and the $p$ corresponding to the inner product. Finally, it outputs the point $p$ maximizing the distance among these candidates.

**Definition 3.1** (Oblivious Algorithm). *Given query $q \in \mathbb{R}^d$, dataset $P \subset \mathbb{R}^d$ of size $n$, and projection vectors $a_1, \ldots, a_N \sim \mathcal{N}(0, I_d)$, the oblivious algorithm computes $|\langle a_i, q - p \rangle|$ for all $i \in [N]$, $p \in P$, selects the $c_N \cdot N$ pairs $(i, p)$ achieving the largest values (for some absolute constant $c_N$), and returns $\arg\max_{p \in S} \|q - p\|$, where $S$ is the set of points with the largest values. This procedure is expected to return a $(1 + \varepsilon)$-AFN with constant probability.*

The goal of this section is to prove the following result.

**Lemma 3.2.** *Consider the dataset $P$ defined above (consisting of $n/2$ copies each of $p_-$ and $p_+$). Let $c_N$ be the constant from Definition 3.1. Suppose $n \geq 2c_N N$ and $\log N \leq d^{0.02}/C$ for a sufficiently large absolute constant $C$. With probability at least $1 - e^{-\Theta(d^{0.02})} - 1/N$ over $a_1, \ldots, a_N$, the oblivious algorithm (Definition 3.1) on query $q$ returns a point $\hat{p}$ with $\|q - \hat{p}\| \leq d^{0.01}$, while $\max_{p \in P} \|q - p\| \geq d^{0.5}$. In particular, $\hat{p}$ is not a $c$-approximate furthest neighbor for any $c \leq d^{0.49}/2$.*

We extend the idea behind the failure of the algorithm's principle to show that the algorithm itself fails as well. Let the dataset $P$ consist of $n/2$ copies of the point $p_- := -\vec{1} = (-1, \ldots, -1)$ and $n/2$ copies of $p_+ := \vec{1} = (1, \ldots, 1)$. Note that $\|p_+ - p_-\|_2 = 2\sqrt{d}$. Let $a_1, \ldots, a_N \sim N(0, I_d)$ denote the sampled vectors of the algorithm and set $v = a_1/\|a_1\|$. Set $q := p_- + xyv = -\vec{1} + xyv$, where $y := \text{sgn}(\langle a_1, p_+ - p_- \rangle)$ and $x > 0$ to be chosen. This is the adversarially constructed query point.

The proofs of lemmas used in this section are in Section C.

**Lemma 3.3.** *Set $x = d^{0.01}$. Provided $\log N \leq d^{0.02}/C$ for a sufficiently large absolute constant $C$, with failure*

*probability $e^{-\Theta(d^{0.02})} + 1/N$, we have the following:*

$$\|q - p_-\| = d^{0.01},$$
$$\|q - p_+\| > d^{0.5}, \text{ and}$$
$$|\langle q - p_-, a_1 \rangle| > |\langle q - p_+, a_i \rangle| \text{ for all } i \in [N].$$

*Proof.* The three failure events are: $F_1 := \{\|a_1\| < \sqrt{d}/2\}$, $F_4 := \{|\langle p_+ - p_-, v \rangle| > d^{0.01}\}$, and $F_C := \{\max_{i > 1} |\langle q - p_+, a_i \rangle| > t(2\sqrt{d} + x)\}$. By Lemmas C.1 and C.4, and Corollary C.6, $\Pr[F_1 \cup F_4 \cup F_C] \leq e^{-\Theta(d)} + e^{-\Theta(d^{0.02})} + 1/N = e^{-\Theta(d^{0.02})} + 1/N$. We work on $\overline{F_1} \cap \overline{F_4} \cap \overline{F_C}$.

**First and second conditions.** $\|q - p_-\| = \|xyv\| = x = d^{0.01}$. Since $x = d^{0.01} < \sqrt{d}$ for all $d \geq 2$, Lemma C.2 gives $\|q - p_+\| > \sqrt{d} = d^{0.5}$.

**Third condition, $i = 1$.** On $\overline{F_4}$, $|\langle p_+ - p_-, v \rangle| \leq d^{0.01} = x$, so $x > \frac{1}{2}|\langle p_+ - p_-, v \rangle|$. Lemma C.3 therefore gives $|\langle q - p_+, a_1 \rangle| < |\langle q - p_-, a_1 \rangle|$.

**Third condition, $i > 1$.** On $\overline{F_1}$, Lemma C.7 gives $|\langle q - p_-, a_1 \rangle| = x\|a_1\| \geq x\sqrt{d}/2 = d^{0.51}/2$. On $\overline{F_C}$, $|\langle q - p_+, a_i \rangle| \leq t(2\sqrt{d} + x) \leq 3t\sqrt{d}$ for $x = d^{0.01} \leq \sqrt{d}$ and large $d$. The condition $d^{0.51}/2 > 3t\sqrt{d}$ simplifies to $d^{0.01} > 6t = \Theta(\sqrt{\log N})$, which holds for $d$ large enough relative to $\log N$. $\square$

*Proof of Lemma 3.2.* By Lemma 3.3, with probability $\geq 1 - e^{-\Theta(d^{0.02})} - 1/N$), we have $\|q - p_-\| = x = d^{0.01}$, $\|q - p_+\| \geq d^{0.49} \cdot d^{0.01} = d^{0.5}$, and $|\langle q - p_-, a_1 \rangle| > |\langle q - p_+, a_i \rangle|$ for all $i \in [N]$. The dataset contains $n/2 \geq c_N N$ identical copies of $p_-$; each paired with $a_1$ yields inner product $x\|a_1\|$, which exceeds every $(a_i, p_+)$ value. Hence all $c_N N$ candidate slots are occupied by copies of $p_-$, and $p_+$ is never added to $S$. The algorithm returns $\hat{p} = p_-$ at distance $d^{0.01}$, a factor $d^{0.49}$ short of the true furthest-neighbor distance $d^{0.5}$. $\square$

## Acknowledgements

The work is partially supported by DARPA expMath, ONR MURI 2024 award on Algorithms, Learning, and Game Theory, Army-Research Laboratory (ARL) Grant W911NF2410052, NSF AF:Small grants 2218678, 2114269, 2347322.

## Impact Statement

This paper presents work whose goal is to advance the field of Machine Learning. There are many potential societal consequences of our work, none which we feel must be specifically highlighted here.

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

# A. Omitted details and proofs in Section 2.1

*Proof of Lemma 2.2.* We utilize the well known standard Gaussian tail bounds (Feller, 1991). For a standard normal variable $Z \sim N(0,1)$ and $x \geq 0$,

$$\frac{Be^{-x^2/2}}{x} \leq \Pr[Z \geq x] \leq \frac{e^{-x^2/2}}{x}. \tag{3}$$

**Proof of Equation** (2)   Let $\Delta' = p' - q$. By assumption, we have $\|\Delta'\| < \|p - q\|\frac{1+\delta}{c}$. The normalized projection $Z = \frac{a \cdot \Delta'}{\|\Delta'\|}$ is distributed as $\mathcal{N}(0,1)$. We analyze the probability that the projection exceeds the threshold $T_{near} = \frac{t\|p-q\|}{c}(1-\delta)$:

$$\Pr_a[a \cdot \Delta' \geq T_{near}] = \Pr\left[Z \geq \frac{T_{near}}{\|\Delta'\|}\right]$$

$$< \Pr\left[Z \geq \frac{T_{near}c}{(1+\delta)\|p-q\|}\right]$$

$$= \Pr\left[Z \geq t(1-\delta)/(1+\delta)\right].$$

Using the upper tail bound from (3) and the definition of $t$, the claim follows.

**Proof of Equation** (1)   Let $\Delta = p - q$. We analyze the probability that the projection exceeds $T_{far} = \frac{t\|p-q\|}{c}(1+\delta)$:

$$\Pr_a[a \cdot \Delta \geq T_{far}] = \Pr\left[Z \geq \frac{T_{far}}{\|p-q\|}\right] = \Pr[Z \geq \lambda],$$

for $\lambda = \frac{t(1+\delta)}{c}$. Using the lower tail bound from (3):

$$\Pr[Z \geq \lambda] \geq \frac{B}{\lambda}e^{-\lambda^2/2} \approx \frac{1}{t}e^{-\lambda^2/2}.$$

To relate this to $n$, we substitute the exponent derived from the definition of $t$, that is,

$$e^{-\lambda^2/2} = \left(e^{-\frac{t^2(1-\delta)^2}{2(1+\delta)^2}}\right)^{\frac{\lambda^2(1+\delta)^2}{t^2(1-\delta)^2}} = \left(\frac{1}{2nt}\right)^{\frac{\lambda^2(1+\delta)^4}{(1-\delta)^2c^2}}.$$

Noting that $(1+\delta)^4/(1+\delta)^2 = (1+O(\delta))$ for $\delta \leq 1/2$ yields the claim. $\qquad\square$

# B. Omitted details and proofs in Section 2.2

**Definition B.1.** *For a set of $n$ points $P \subset \mathbb{R}^d$, we define the box width of $P$ as*

$$bw(P) = \max_{i=1,\ldots,d}|\max_{p \in P}p_i - \min_{p \in P}p_i|,$$

*and define the center of $P$ as $ct(P)$ such that*

$$ct(P)_i = \frac{1}{2}(\max_{p \in P}p_i + \min_{p \in P}p_i),$$

*where $p_i$ is the $i$-th coordinate of $p$.*

**Lemma B.2.** *Given a set of $n$ points $P \subset \mathbb{R}^d$. For any $p \in P$ and any query $q \in \mathbb{R}^d$, we have*

1. *$\|p - q\| \geq \|q - ct(P)\| - \frac{\sqrt{d}}{2}bw(P)$,*

2. *$\|p - q\| \leq \|q - ct(P)\| + \frac{\sqrt{d}}{2}bw(P)$.*

*Proof.* For any $p \in P$, we have $|p_i - ct(P)_i| \leq \frac{1}{2}|\max_{p \in P} p_i - \min_{p \in P} p_i| \leq \frac{1}{2}bw(P)$. So we get $\|p - ct(P)\| = \sqrt{\sum_{i=1}^d |p_i - ct(P)_i|^2} \leq \frac{\sqrt{d}}{2}bw(P)$. By the triangle inequality, we have

$$\|p - q\| \leq \|q - ct(P)\| + \|p - ct(P)\|$$
$$\leq \|q - ct(P)\| + \frac{\sqrt{d}}{2}bw(P)$$

and

$$\|p - q\| \geq \|q - ct(P)\| - \|p - ct(P)\|$$
$$\geq \|q - ct(P)\| - \frac{\sqrt{d}}{2}bw(P).$$

$\square$

*Proof of Lemma 2.6.* By Lemma B.2, we have

1. $\min_{p' \in P} \|p' - q\| \geq \|q - ct(P)\| - \frac{\sqrt{d}}{2}bw(P)$,

2. $\max_{p \in P} \|p - q\| \leq \|q - ct(P)\| + \frac{\sqrt{d}}{2}bw(P)$.

Since $\|q - ct(P)\| \geq \frac{(1+c)\sqrt{d}}{2(c-1)}bw(P)$, then we have

$$\frac{\min_{p' \in P} \|p' - q\|}{\max_{p \in P} \|p - q\|} \geq \frac{\|q - ct(P)\| - \frac{\sqrt{d}}{2}bw(P)}{\|q - ct(P)\| + \frac{\sqrt{d}}{2}bw(P)} \geq \frac{1}{c}.$$

$\square$

**Definition B.3.** *Given a parameter $\eta > 0$ and a feasible set $\mathcal{X} \subset \mathbb{R}^d$. Define the grid covering over $\mathcal{X}$ with spacing $\eta$ as*

$$G_{\eta, \mathcal{X}} = \{g \in \mathcal{X} | g = \eta \cdot x, x \in \mathbb{Z}^d\}.$$

**Lemma B.4.** *Given a parameter $\eta > 0$ and a ball $B = B(0, r) \subset \mathbb{R}^d$. Let $G_{\eta, B}$ be the grid covering over $B$ with spacing $\eta$. For any $q \in B$, there exists a point $g \in G_{\eta, B}$ such that $\|g - q\| \leq \sqrt{d}\eta$. Moreover, its size is bounded by*

$$|G_{\eta, B}| \leq \left(\frac{2R}{\eta} + 1\right)^d.$$

*Proof.* Let $q = (q_1, \ldots, q_d)$. We find $g = \eta \cdot x$ as follows:

$$x_i = \begin{cases} \lfloor q_i/\eta \rfloor & \text{if } q_i \geq 0 \\ \lceil q_i/\eta \rceil & \text{if } q_i < 0 \end{cases}.$$

We verify the properties of $g$ as follows: First, we have $\eta \cdot |x_i| \leq |q_i|$. So $\|g\| \leq \|q\| \leq r$, which implies $g \in B$. Next, we have $|\eta \cdot x_i - q_i| \leq \eta$. Thus, $\|g - q\| \leq \sqrt{d}\eta$.

It remains to prove the cardinality bound. For any grid point $g = \eta \cdot x \in G_{\eta, B}$, we have

$$\|g\| \leq R \quad \Rightarrow \quad |x_i| \leq \frac{R}{\eta} \quad \text{for all } i \in [d].$$

Thus each coordinate $x_i$ can take at most $\lfloor 2R/\eta \rfloor + 1$ integer values. By independence across dimensions,

$$|G_{\eta, B(0,R)}| \leq (\lfloor 2R/\eta \rfloor + 1)^d \leq \left(\frac{2R}{\eta} + 1\right)^d.$$

$\square$

## C. Omitted Details from Section 3

**Lemma C.1.** $\Pr\left[\|a_1\| \le \sqrt{d}/2\right] < e^{-\Theta(d)}$.

*Proof of Lemma C.1.* We have $\|a_1\|^2 \sim \chi^2(d)$ with $\mathbb{E}[\|a_1\|^2] = d$. By the Chernoff bound via the moment generating function: for any $s > 0$,

$$\Pr\left[\|a_1\|^2 \le \tfrac{d}{4}\right] \le e^{sd/4} \mathbb{E}\left[e^{-s\|a_1\|^2}\right] = e^{sd/4}(1 + 2s)^{-d/2}.$$

Setting $s = 3/2$, the bound becomes $e^{3d/8} \cdot 4^{-d/2} = e^{-d(\ln 2 - 3/8)}$. Since $\ln 2 - 3/8 > 0.31 > 0$, we get $\Pr[\|a_1\| \le \sqrt{d}/2] \le e^{-\Theta(d)}$. $\square$

**Lemma C.2.** *If* $x < \sqrt{d}$ *then* $\|q - p_+\| > \sqrt{d}$.

*Proof of Lemma C.2.* We compute $q - p_+ = xyv - 2 \cdot \vec{1}$, so $\|q - p_+\|^2 = x^2 \|v\|^2 - 4xy \left\langle v, \vec{1} \right\rangle + 4\left\|\vec{1}\right\|^2 = x^2 - 4xy \left\langle v, \vec{1} \right\rangle + 4d$. Since $v = a_1/\|a_1\|$ and $y = \text{sgn}(\langle a_1, \vec{1}\rangle) = \text{sgn}(\langle v, \vec{1}\rangle)$, we have $y\left\langle v, \vec{1} \right\rangle = \left|\left\langle v, \vec{1} \right\rangle\right| \ge 0$. By Cauchy-Schwarz, $\left|\left\langle v, \vec{1} \right\rangle\right| \le \|v\| \left\|\vec{1}\right\| = \sqrt{d}$, so

$$\|q - p_+\|^2 \ge x^2 - 4x\sqrt{d} + 4d = (2\sqrt{d} - x)^2.$$

For $x < \sqrt{d}$, $(2\sqrt{d} - x)^2 > d$, hence $\|q - p_+\| > \sqrt{d}$. $\square$

**Lemma C.3.** *If* $x > \frac{1}{2} |\langle p_+ - p_-, v\rangle|$ *then* $|\langle q - p_+, a_1\rangle| < |\langle q - p_-, a_1\rangle|$.

*Proof of Lemma C.3.* Since $v = a_1/\|a_1\|$, we have $\langle v, a_1\rangle = \|a_1\|$, giving

$$\langle q - p_-, a_1\rangle = \langle xyv, a_1\rangle = xy \|a_1\|,$$
$$\langle q - p_+, a_1\rangle = xy \|a_1\| - 2\left\langle \vec{1}, a_1 \right\rangle = \|a_1\| \left(xy - 2\left\langle \vec{1}, v \right\rangle\right).$$

So $|\langle q - p_-, a_1\rangle| = x \|a_1\|$ and it suffices to show $\left|xy - 2\left\langle \vec{1}, v \right\rangle\right| < x$. Set $u := xy$ and $w := 2\left\langle \vec{1}, v \right\rangle$. By algebra, $|u - w|^2 = u^2 - w(2u - w)$, so $|u - w| < |u|$ is equivalent to $w(2u - w) > 0$. Since $y = \text{sgn}(\langle v, \vec{1}\rangle)$, $u$ and $w$ have the same sign, so $w(2u - w) > 0$ is equivalent to $2|u| > |w|$. The hypothesis gives $|w| = |\langle p_+ - p_-, v\rangle| < 2x = 2|u|$. $\square$

**Lemma C.4.** $\Pr\left[|\langle p_+ - p_-, v\rangle| > d^{0.01}\right] \le e^{-\Theta(d^{0.02})}$.

*Proof.* Since $v = a_1/\|a_1\|$, the vector $v$ is uniform on $\mathbb{S}^{d-1}$. By Lévy's lemma, for any unit vector $e$ and $\epsilon \in (0, 1)$, $\Pr[|\langle e, v\rangle| > \epsilon] \le 2e^{-d\epsilon^2/2}$. Let $\hat{1} := \vec{1}/\sqrt{d}$, so $\langle p_+ - p_-, v\rangle = 2\left\langle \vec{1}, v \right\rangle = 2\sqrt{d} \langle \hat{1}, v\rangle$. Then

$$\Pr\left[|\langle p_+ - p_-, v\rangle| > d^{0.01}\right] = \Pr\left[|\langle \hat{1}, v\rangle| > \tfrac{1}{2}d^{-0.49}\right]$$
$$\le 2\exp\left(-\frac{d \cdot d^{-0.98}}{8}\right)$$
$$= 2e^{-d^{0.02}/8} = e^{-\Theta(d^{0.02})}.$$

$\square$

**Lemma C.5.** *There exists* $t \in \Theta(\sqrt{\log N})$ *such that for any* $i > 1$,

$$\Pr\left[|\langle q - p_+, a_i\rangle| > t(2\sqrt{d} + x)\right] < 1/N^2.$$

*Proof.* For $i > 1$, $a_i \sim \mathcal{N}(0, I_d)$ is independent of $q = -\vec{1} + xyv$, which depends only on $a_1$. Conditioned on $q$, $\langle q - p_+, a_i \rangle \sim \mathcal{N}(0, \|q - p_+\|^2)$. By the triangle inequality, $\|q - p_+\| = \left\| xyv - 2\vec{1} \right\| \leq x + 2\sqrt{d}$, so the threshold normalized by $\|q - p_+\|$ satisfies $t(2\sqrt{d} + x)/\|q - p_+\| \geq t$. By the Gaussian tail bound,

$$\Pr\left[ |\langle q - p_+, a_i \rangle| > t(2\sqrt{d} + x) \right] \leq \Pr[|Z| > t] \leq 2e^{-t^2/2},$$

for $Z \sim \mathcal{N}(0, 1)$. Choosing $t = C\sqrt{\log N}$ with constant $C > 2$ gives $2e^{-t^2/2} = 2N^{-C^2/2} < N^{-2} = 1/N^2$ for large enough $N$. $\square$

**Corollary C.6.**

$$\Pr\left[ \max_{i>1} |\langle q - p_+, a_i \rangle| > t(2\sqrt{d} + x) \right] < 1/N.$$

*Proof.* By a union bound over $i = 2, \ldots, N$:

$$\Pr\left[ \max_{i>1} |\langle q - p_+, a_i \rangle| > t(2\sqrt{d} + x) \right] \leq N \cdot \frac{1}{N^2} = \frac{1}{N}. \qquad \square$$

**Lemma C.7.** $|\langle q - p_-, a_1 \rangle| = x \|a_1\|$.

*Proof.* We have $q - p_- = -\vec{1} + xyv - (-\vec{1}) = xyv$, so $\langle q - p_-, a_1 \rangle = xy \langle v, a_1 \rangle = xy \|a_1\|$, using $\langle v, a_1 \rangle = \|a_1\|$. Taking absolute values: $|\langle q - p_-, a_1 \rangle| = x \|a_1\|$ since $x > 0$ and $|y| = 1$. $\square$

