# OpenReview forum: "Adversarially Robust Approximate Furthest Neighbor"
_ICML.cc/2026/Conference — ICML 2026 regular_

### Official Review · Reviewer_ALh3 · 2026-03-11

**Soundness:** 4
**Presentation:** 4
**Significance:** 3
**Originality:** 3
**Overall Recommendation:** 5
**Confidence:** 4

**Summary:**

This paper designs an algorithm and data structure for adversarially robust approximate furthest neighbor search. The model considered is one where an adversary provides query points and the algorithm should produce a point from the original dataset whose distance is at least some constant times the maximum distance from the query. The adversary can have access to the previous answers of the algorithm, as well as its internal randomness. The algorithm the authors claim achieves this goal with high probability against any adversary by carefully modifying inner-product-preserving random projections and via a suitably designed covering over the dataset space. Robustness is guaranteed via a union bound and query efficiency is achieved by utilizing a distance estimator algorithm.

**Compliance With Llm Reviewing Policy:**

Affirmed.

**Final Justification:**

The paper is a solid theoretical contribution for an important problem. It can improve a few things presentation-wise (perhaps add a small discussion on regimes where the space-complexity blow-up is not ideal), but I overall I believe it presents a valuable contribution to the field.

**Key Questions For Authors:**

Would approaches utilizing privacy work for this problem? Is this a possible direction that can help is some regimes?

**Limitations:**

Yes

**Strengths And Weaknesses:**

1. **Significance**: The problem considered is approximate furthest neighbor search. A close relative to the classic ANN problem, this problem arises in many situations in machine learning and the physical sciences, so studying it is important. As is standard with query+search problems on efficient data structures, the model considered is one where the algorithm processes the input dataset first and then receives queries to which they should respond. The twist now is that the queries are received adaptively and an adversary aims to break the algorithm's correctness by producing a query on the "edge" of the algorithm's randomness. Such attacks are popular in the big-data community in the last few years, as designing algorithms that work despite them is significantly harder. The paper's contribution is the solution of this problem: the authors claim an algorithm whose runtime and space complexity almost match the "oblivious" algorithm. This is quite significant as it departs from the commonly held (though increasingly eroded) conception that adversarially robust algorithms must spend a lot more time and space to protect themselves from the adversary.

2. **Originality**: The algorithm builds on top of several existing ingredients (like inner-product-preserving projections and distance approximating data structures). These ingredients are modified and consumed in a natural way to facilitate the program intended by the authors.
   * One of the conceptual contributions is the introduction of "good" query points with respect to projections: points where the projection roughly maintains the relationship between the furthest neighbor and the remaining points. It is a valuable insight that Gaussian random projections achieve this property with high probability even when a "slack" is required around the query point. This quickly gives an oblivious algorithm for retrieving furthest neighbors. Though this algorithm is a seemingly "direct" consequence of existing random projection theory, the additional requirement of "goodness" (that later helps in robustness) and the lack of asymptotic degradation to achieve it is important.
   * Now that furthest neighbors can be achieved obliviously with the algorithm also guaranteeing "goodness" with constant probability for a fixed query, the authors argue for robustness. This is done via a standard covering+union bound algorithm that has been used in prior robustification papers. First, they only answer queries within a certain (large enough) radius from the centroid of the dataset. For the points within the ball, the algorithm lays a grid of good points and uses enough random projections to ensure that all good points are "seen". One issue with such arguments is that they typically scale the space complexity by a factor of the dimension (log of $2^d$). Indeed, the paper defines $k = \log(n|Q|)$ and $|Q| = \Omega(2^d)$, so this has the unfortunate consequence of adding another factor of $d$ in the space complexity. Approaches utilizing differential privacy for ANN, for instance, do not suffer from this blow-up, though they scale by the root of the number of queries. Additionally, the existing work on robust ANN that uses "for-all" grid arguments also suffers from this scalling.
   * The algorithm is wrapped up via a clever usage of an existing robust distance estimation framework. Without this crucial step, query complexity would be suboptimal.

Overall, the algorithm presented makes solid use of existing technical tools and proposes interesting structural insights for the furthest nearest neighbor problem. The space complexity does suffer a known problem in the high-dimensional regime, though the authors have optimized the remaining parts of the algorithm to match the oblivious complexity.


3. **Presentation**: The presentation is very nice - the paper provides solid mathematical definitions and flows very well.

4. **Soundness**: The derivation and analysis seem correct to me.

---

> ### Author Rebuttal · Authors · 2026-03-31
>
> Thank you for your thorough review. We are encouraged by your positive comments, including the solid use of existing technical tools and the appreciation of our structural insights for the furthest nearest neighbor problem. We address your question below.
>
> > Would approaches utilizing privacy work for this problem? Is this a possible direction that can help in some regimes?
>
> As you noted, differential privacy (DP) approaches often scale with $\sqrt{Q}$. In comparison, our robustification approach incurs a $d$-factor. Assuming a DP approach works, this suggests it might be preferable in regimes with $Q = O(d^2)$, neglecting refreshing of data structures or the requirement that internal randomness need not be revealed.
>
> However, we are not aware of any DP approaches for this problem; it seemed more amenable to the robustification approach presented. Regarding the feasibility of a DP approach, since other problems admit both approaches (DP and “for-all” guarantee), it is likely that AFN admits both approaches too. One perhaps may use the `slack’ provided by the (analysis of the) base data structure to aggregate answers with added noise. Overall, it would be a nice direction that would complement the understanding of this problem.

---

> > ### Author Rebuttal · Reviewer_ALh3 · 2026-04-02
> >
> > Thank you for your response. I will maintain my score.

---

### Official Review · Reviewer_nDF7 · 2026-03-11

**Soundness:** 3
**Presentation:** 3
**Significance:** 2
**Originality:** 2
**Overall Recommendation:** 3
**Confidence:** 4

**Summary:**

This paper studies the approximate furthest neighbor (AFN) problem in the adaptive query setting. The authors propose a data structure that attains the query-time complexity targeted by prior theoretical work. The main technical idea is to use the information of multiple data-structure copies and an additional distance-estimation component. Overall, this paper is primarily a theoretical contribution.

**Compliance With Llm Reviewing Policy:**

Affirmed.

**Final Justification:**

This paper has no experimental results, although the authors claim its practical utility. I also have a serious concern about the practical feasibility of using multiple copies of the original data structure. Moreover, it lacks a comparison with quantization-based approaches, which may be both space-efficient and highly performant. Therefore, its practical utility remains unverified. If I focus only on the theoretical contribution, it relies heavily on a combination of existing techniques. Although this combination is non-trivial, the overall novelty is somewhat limited for a theory paper, partly because AFN appears to have limited practical relevance and broader impact. Therefore, I will maintain my score. Since other reviewers are positive about this paper, it may be best left to the AC for a final decision.

**Key Questions For Authors:**

None

**Limitations:**

Yes

**Strengths And Weaknesses:**

**Strengths**

(1) The presentation is clear and easy to follow.

(2) The theoretical analysis seems reasonable and sound.

**Weaknesses**

(1) There are no experiments to validate the effectiveness of the proposed data structure.

(2) From a theoretical perspective, the technical depth is somewhat limited, as the approach appears to be a combination of existing techniques and ideas.

(3) The lower query complexity seems to be achieved at the expense of significantly higher space costs and preprocessing overhead by using $k$ copies of the data structure. Its practical utility is therefore questionable.

---

> ### Author Rebuttal · Authors · 2026-03-31
>
> Thank you for your valuable feedback! We are glad that the clarity of presentation and the theoretical soundness of our paper are valued. We address the main weaknesses below.
>
> > **Weakness (1)**: Lack of experiments
>
> We focus on proving novel and rigorous theoretical bounds for robust furthest neighbor, which we believe is of interest to the broader ICML community. However, we would like to point out that constant factors in our algorithms are quite small and the algorithm is quite clean. As a proof of concept, we implemented our algorithm and compared it to the obviously robust baseline of doing a linear scan (note we are not aware of any other robust algorithms with sublinear in $n$ query time).
>
> The following table below demonstrates the speedup over the linear scan (whose runtime is normalized to 1) for different values of $c$. Our robust data structure is faster than the naive linear scan while being provably robust against adaptive queries.
>
> | Algorithm | $c = 2.5$ | $c = 3.5$ | $c = 5$ | $c = 10$ |
> | :--- | :--- | :--- | :--- | :--- |
> | **Ours** | 100% 1.85x | 99.2% 10.5x | 98.3% 29x | 88.2%  33.2x |
>
> **Table 1**: Algorithm performance with $n=10^5$ and $d=100$ with random points on a hypersphere and $100$ random queries. Each entry shows approximation quality (as %) and avg. speedup over linear scan.
>
> To summarize, while our main focus is proving theoretical bounds, we believe our algorithm should also have strong practical performance as demonstrated above, and we leave a detailed empirical study as an interesting direction for future work.
>
> > **Weakness (2)**: From a theoretical perspective, the technical depth is somewhat limited, as the approach appears to be a combination of existing techniques and ideas.
>
> We do make use of existing techniques, although the combination thereof is non-trivial. We provide an overview of our techniques in Section 1.1, including robustifying existing data structures while obtaining a matching query complexity as the non-robust setting, and a comparison with prior approaches. There also appears to be a consensus among other reviewers on this (please, see quotes below).
>
> - *While the latter part of their approach is a standard trick + known algorithm, combining all the things and showing it works even in adaptive setting is a significant contribution* (GdP5) \
> *The techniques are non-trivial and refreshing [...] The authors provide a generic technique for how to achieve adaptivity [...]* (SYtA)\
> *Overall, the algorithm presented makes solid use of existing technical tools and proposes interesting structural insights for the furthest nearest neighbor problem.* (ALh3)
>
> > **Weakness (3)**: The lower query complexity seems to be achieved at the expense of significantly higher space costs and preprocessing overhead by using copies of the data structure. Its practical utility is therefore questionable.
>
> We would like to clarify the relationship between query complexity, preprocessing, and space usage in our algorithm.
>
> **Space:** Our approach does not asymptotically increase space usage. The way we achieve a lower query complexity is described in point (3.) of Section 1.1. To summarize that, we achieve a lower query complexity by combining two ingredients: (i) querying only $O(\log n)$ base data structures and (ii) using a distance estimator framework. None of these two approaches incurs a higher space cost than that already required by the furthest neighbor data structures described in points (1.) and (2.) of Section 1.1. We also note that our space consumption is in line with that of other robust algorithms (Cherapanamjeri & Nelson, 2020; Bateni et al., 2024; Andoni et al., 2026). Since they all exhibit a $d^2$-factor dependency, breaking this barrier or proving it necessary is an interesting direction for future work.
>
> **Preprocessing vs Query Time:** In any setting where the number of queries is at least $\Omega(d)$, the preprocessing overhead is fully offset by the faster query response. Recall from the explanation above that we (i) query only $O(\log n)$ base data structures and (ii) use a distance estimator framework. Step (i) does not incur preprocessing overhead. Step (ii) is where we incur a $d$-factor preprocessing overhead, which is precisely what enables a corresponding $d$-factor improvement at query time. Therefore, when there are at least $d$ queries, the amortized preprocessing time is asymptotically the same as that we would get without query optimization.
>
> **Utility:** Having established that our preprocessing does not necessarily hinder utility, we view assessing its practicality as a valuable direction for future work. We also hope our techniques may find other theoretical applications and inform future practical works.
>
> Finally, if you feel that our response has adequately addressed your major concerns, we would appreciate it if you could possibly adjust your score accordingly.

---

> > ### Author Rebuttal · Reviewer_nDF7 · 2026-04-03
> >
> > Thank you for your response. AFN is a more specialized problem than ANNS, so I would expect either stronger technical novelty or more compelling practical justification. (1) Regarding novelty, I still feel that the contribution is not sufficiently strong for a primarily theoretical paper. (2) I also remain unconvinced about the practical utility, mainly due to the space-time tradeoff. The improved query bound comes at the cost of substantial space and preprocessing overhead, and the overall overhead still appear very heavy. This tradeoff becomes particularly unattractive when $c$ is close to 1, which is often the practically important regime in high-dimensional settings.

---

> > > ### Author Response · Authors · 2026-04-03
> > >
> > > Thank you for the feedback!
> > >
> > > (1) We would like to point out that there are still several improvements that we achieve that are not yet possible for robust ANN:
> > >
> > > - Dependence on aspect ratio: All robust ANN approaches we are aware of (e.g. [1]) have a dependence on log(aspect ratio) of the dataset, which we avoid by exploring specific properties of the AFN problem. Note that [1] only presents the (c,r)- ANN problem, but they sill need to binary search over all possible values of ‘r’ leading to a log(aspect ratio) dependency.
> > >
> > > - No d factor in the query time: Traditionally in geometric algorithms, we can always take d = O(log n) via the JL lemma. However, this is not possible in the adversarially robust case as the JL matrix cannot guarantee norm preservation against adversarially chosen inputs. Consequently, existing robust ANN algorithms have a d multiplicative factor in the query time compared to their non-robust counterparts. In contrast, we have no such d multiplicative factor and actually *match* the query time of the non-robust baseline (up to log(n) factors), through two ideas: sampling O(log n) base data structures using fresh randomness and then recursively using the adaptive data structure of Cherapanamjeri and Nelson.
> > >
> > > - **Sublinear** in n space: For $c \ge \sqrt{2}$ approximation, our space is actually *sublinear* in n. For ANN, every robust approach that we are aware of (e.g. based on LSH), always require Omega(n) space for all factor c approximation.
> > >
> > > [1] Efficient Algorithms for Adversarially Robust Approximate Nearest Neighbor Search. Andoni, Haris, Kelman, Onak.
> > >
> > >
> > > (2) Indeed when c-> 1, a trivial linear scan would be preferable, but this is also true for the best known non-robust algorithm (for AFN and ANN). However for any c > 1, we get strictly sublinear query time in n which is preferable for large-scale datasets where n is often several order of magnitudes larger than d. Indeed, we could scale our experiments to large n where a linear in n scan is too prohibitive.
> > >
> > > Lastly, we would like to point out that in practice, the query time is also a major concern and we match that of the best non-robust algorithm. Indeed, in [2] for a production level ANN data structure, the pre-processing step is allowed to take *multiple days* to build (e.g. [2] mentions that one version of their data structure took 5 days to build), whereas the query-time is measured in *milliseconds.* This shows that in some practical scenarios, we often allow for large preprocessing with the trade-off of fast queries. This is the setting where we believe our method would be very practical.
> > >
> > > [2] DiskANN: Fast Accurate Billion-point Nearest Neighbor Search on a Single Node. Subramanya, Devvrit, Kadekodi, Krishaswamy, Simhadri.

---

### Official Review · Reviewer_SYtA · 2026-03-12

**Soundness:** 4
**Presentation:** 4
**Significance:** 4
**Originality:** 4
**Overall Recommendation:** 5
**Confidence:** 4

**Summary:**

The paper provides a theoretical construction for an adaptive approximate furthest neighbor problem. In the furthest neighbour problem, one wishes to construct a data structure that takes a set of points in R^d and answers queries of the form "what is the input point farthest from q?". In the approximate version, one does not necessarily need to return the exact farthest point, but is satisfied with returning an input point whose distance to the query point q is at least 1/c times the farthest distance. This primitive is related to the "nearest neighbor" version (which is fundamental) and has arisen in prominence lately in various ML contexts. The current paper considers the problem in a new modern context, that of adversarial queries: namely, the user interacting with the data structure is now assumed to have malicious intent and is allowed to adapt its queries based on the history. The goal is to force the algorithm to take a long time. An algorithm that maintains its guarantees even in the presence of such an adversary is called adversarially robust. The authors provide such a construction in the paper.

**Compliance With Llm Reviewing Policy:**

Affirmed.

**Key Questions For Authors:**

none

**Limitations:**

yes

**Strengths And Weaknesses:**

The paper is a solid contribution to a fundamental problem. The techniques are non-trivial and refreshing, and match the SOTA for the non-oblivious case (so being adversarially robust comes at no extra asymptotic cost). Obtaining adversarially robust algorithms is an important concern, and the contribution of the paper goes beyond the application at hand (the farthest neighbor) - the authors provide a generic technique for how to achieve adaptivity and it's likely that the techniques will transfer to other algorithms as well. The paper is coherent and precise.

---

> ### Author Rebuttal · Authors · 2026-03-31
>
> Thank you for your valuable feedback and for the time spent reviewing our paper. We are encouraged that our problem is viewed as fundamental and the techniques as refreshing.

---

> > ### Author Rebuttal · Reviewer_SYtA · 2026-04-05
> >
> > I had no questions.

---

### Official Review · Reviewer_GdP5 · 2026-03-13

**Soundness:** 3
**Presentation:** 3
**Significance:** 3
**Originality:** 2
**Overall Recommendation:** 4
**Confidence:** 3

**Summary:**

The paper considers the problem of approximate furthest neighbor where queries are made by an adaptive adversary. Given a set of points $P$ and query $q_i$ at time $i$, the  task is to return point furthest away from $q_i$ in $P$, and the query $q_i$ may depend on outputs for previous queries $\{q_1 ,\cdots, q_{i-1}\}$. This work gives a data structure that in time $O(n^{1/c^2} + d)$ returns a $c$-approximate furthest neighbor, matching the previous known bounds for a non-adaptive adversary.

**Compliance With Llm Reviewing Policy:**

Affirmed.

**Final Justification:**

Good set of results and technical contribution, I maintain my positive evaluation of the work and lean towards acceptance.

**Key Questions For Authors:**

No clarifying questions for the authors.

**Limitations:**

Yes.

**Strengths And Weaknesses:**

The paper is well written and clearly motivated. The authors show that even in case of adaptive adversary, it is possible to obtain same runtime bounds as an oblivious adversary. The technical contribution lies in the data structure, where they first argue that if the data structure is "good" for a point $q$, it is also good for any point close enough to $q$. Then using multiple copies of such data structure, along with robust distance estimation from previous work, they show how to efficiently select a point that is guaranteed to be good approximate solution. While the latter part of their approach is a standard trick + known algorithm, combining all the things and showing it works even in adaptive setting is a significant contribution, and they argue that it even works in harder setting where the adversary even gets to observe the internal randomness of the algorithm. The work is primarily theoretical, no experiments are provided.

---

> ### Author Rebuttal · Authors · 2026-03-31
>
> Thank you for your valuable feedback and for the time spent reviewing our paper. We are glad that you find our contribution significant.

---

> > ### Author Rebuttal · Reviewer_GdP5 · 2026-04-02
> >
> > All relevant concerns are addressed.

---

### Decision · Program_Chairs · 2026-04-30

**Decision:**

Accept (regular)

**Comment:**

The paper studies approximate furthest neighbor search under adversarial queries, extending the classical setting to allow queries that depend on responses to previous queries. The authors construct a data structure that matches the best known query-time guarantees from the oblivious setting while maintaining correctness under this stronger adversarial model. The approach combines standard geometric tools such as random projections, distance estimation, and covering/union-bound arguments in a careful and unified way.

Reviewers agree the work is technically sound and clearly presented. While the technical components are largely known, their combination in the adaptive setting is non-trivial and yields a clean robustness guarantee without asymptotic degradation. The main concerns are the absence of empirical evaluation and somewhat limited novelty beyond the composition of existing techniques, along with possible space overhead due to multiple data structure copies. However, adversarial robustness is not necessarily a feature demonstrated with experiments and it has been an increasingly relevant topic in machine learning. Thus, the paper is a good fit for ICML.